# Sharing-Economy Ecosystem: A Comprehensive Review and Future Research Directions

**Samar Abdalla ***, **Joseph Amankwah-Amoah *** and **Amgad Badewi**

Department of Management, Kent Business School, University of Kent, Kent ME4 4TE, UK
* Correspondence: sa2009@kent.ac.uk (S.A.); j.amankwah-amoah@kent.ac.uk (J.A.-A.)

**Abstract:** This research study reviews the literature on the participants in the sharing economy (SE) ecosystem and its impact on the participants, creating and capturing value through increasing the understanding of the ecosystem's novel models. The review classifies the participants in the SE ecosystem into primary and secondary participants. The classification is based on the connection to the core network/ecosystem and the role of the participants in the ecosystem. The primary participants are subdivided into groups: customers are subdivided into New Customers (NC) and Current Customers (CC); providers into Product Providers (PP) and Service Providers (SP); and mediators are subdivided into Small and Medium Mediators (SMM) and Large Mediators (LM). The secondary participants are governments sub-grouped into Local Governments (NG) and National Governments (NG); Theories and methodologies within the academic literature on the sharing economy ecosystem are also examined. The study also analyses the influence of digital sharing and explores the value of digital technologies in management strategies and the value of the integration between participants of SE businesses. Recommended future research directions are outlined according to the conducted review.

**Keywords:** sharing economy; digital economy; participants; ecosystem; digital business model; sustainability

## 1. Introduction

SE is a new phenomenon that is growing around the world. Based on a PWC (PricewaterhouseCoopers) report on SE, the related business will grow by 2133% in 12 years [1]. In addition, previous studies identified a critical pathway for the participants in the sharing economy ecosystem. However, to understand the integration between them and the value that has impacted such integration, there is a need to conduct a conceptual paper to have a clear vision of the SE ecosystem and the participants' role within it. Leung et al. [2] identified the different participants in the SE ecosystem. Although previous studies on SE have provided valuable insights into its ecosystem and relevance to business disciplines [3], they have not specified a particular group of participants, such as customers. Instead, they have been presented as one group without indicating whether they are new or old customers. As a result, they have not been identified as critical elements in the various sharing businesses, distinguished by their specific economic, social, and environmental perspectives [4]. However, understanding the perspectives of a level of subgroups provides the critical insight needed to enrich the investigations into the participants and the integration between them in the SE ecosystem. Moreover, the value of each participant has not been previously explained, even though understanding the importance of the development of SE platforms is critical to achieving sustainable growth.

SE results in more sustainability within the sharing economy [5]. Digital technologies have been recognised as critical tools in SE. Sutheland and Jararhi [6] examined technology as a valuable tool for SE ecosystems. However, its impact on the integration between participants has not been examined in previous studies, which focus on how the technology used in one field may differ from that used in another on SE platforms which assured that

the technologies play an essential role in the SE ecosystem. Therefore, it is vital to identify how digital technologies can impact the social and environmental aspects despite the absence of trust. In addition, Laurell and Sandström [7] discuss market logic and its impact on business. Furthermore, many studies consider the technological impact perspectives, providing a better conceptualisation of SE, but without giving a clear understanding of the value of the technological impact on business sustainability due to the integration between the participants [8,9]. In some studies, digital technology is described as a leading factor in developing business [10], while other researchers dispute this, seeing it as a platform [11,12].

Many articles have also discussed SE-related topics, such as previously mentioned digital technologies in the SE ecosystem and how SE is presented and used through digital channels [13]. The debate around the participants in SE has led the researcher in this study to highlight the value of technology in participants' integration and sustainability. Many studies have examined particular participants and analysed them. As a result, there has been an increase in the awareness and understanding of ecosystems; creating and capturing value through novel models was the focus of attention [14]. However, these studies do not understand the value of the participant's role. For example, is each participant in the ecosystem playing the same role, or do they play primary and secondary roles within it? Accordingly, the research has two objectives. The first objective is to understand the value of technologies in integrating SE participants and their impact on business sustainability. The second objective is to clarify the role of the participants in the SE ecosystem by subdividing them into groups and developing a framework to explain the primary and secondary roles they play in each group.

The research contributes to the sharing economy (SE) ecosystem literature. It first adds to the debate about SE participants by explaining operationalisation and integrating multiple participants in economically viable and sustainable urban reconfigurations in shared mobility. The review contributes to the literature by classifying the participants in the SE ecosystem into primary and secondary groups based on their role within the ecosystem, as detailed in the reviewed articles, which specify the role and the value of each participant in the SE ecosystem. Second, the review contributes to the literature by subdividing the participants in the SE ecosystem into groups. For example, customers are subdivided into New Customers (NC) and Current Customers (CC), providers into Product Providers (PP) and Service Providers (SP), and government into Local Government (NG) and National Government (NG). Furthermore, Mediators are subdivided into Small and Medium Mediators (SMM) and Large Mediators (LM). Nevertheless, it helps to highlight the gap in previous studies that presented participants in general terms. For example, customers in the reviewed studies are not specified as new or current. In addition, previous studies do not specify between Local and National Governments. Second, such subdivisions contribute to the literature by further understanding the participants considered in previous studies. Third, the research presents a framework for understanding the value of technologies in integrating SE participants and business sustainability. It also highlights the primary and secondary participants in the SE ecosystem. Fourth, the research develops a more robust understanding of the potential roles of technology in improving the integration between the participants in the SE ecosystem. In practical terms, the study develops an integrative model that can be used by private and public managers for potential expectations for understanding participants' performance in the SE business. Furthermore, the study explains the value of integration over the short and long term for the sustainability of businesses.

## 2. Literature Review

This section aims to investigate the participants' role in the SE ecosystem. The study starts with a search for a comprehensive definition of SE. Then, it extends the search so that ecosystem definitions and participants in the SE ecosystem can be identified. The following section aims to establish definitions of the SE ecosystem participants. These are examined

through an intensive review of the various definitions of the terms "SE" and "ecosystem," together with that of the participants.

*Who Are the Participants in the SE Ecosystem?*

Starting by considering the term SE, which has many definitions, it was found that various terms are related to it. Dredge and Gyimóthy [15] found 17 such terms, including collaboration consumption, peer-to-peer, and digital economy. Some of these definitions were based on the researchers' understanding of the meaning of SE, which could contradict each other, as they represent the points of view of the different authors. As Görög [16] states, "Although sharing economy phenomenon is clear, it has no bright understanding between academics and practitioners too" [16] (p. 176). The points of view will probably not come together in one definition [17] (p. 1). One definition could be related to the economic system: "a comprehensive definition for the SE, an economic system in which an online platform connects the supply and demand sides to facilitate transactions of giving temporary access to idle resources" [2] (p. 45). However, the explanation provided by Lim [5] of the development of the sharing concept combined with the timeline starting from 1900. In the 1950s, the selling concept was introduced, which developed into marketing in the 1960s. The 1970s was the start of the social market, while the 1980s to the 2000s saw the collaboration concept. Lim's [5] definition was more precise: "The sharing economy is a marketplace that consists of entities (e.g., consumers, organisations) that innovatively and sustainably shape how marketing exchanges of valuable products and resources are produced and consumed through sharing, which can occur when entities take part in (e.g., divide and distribute) the actual or life-cycle use of a product or resource and communicate some form of information, which can be scaled using technology". [5] (p. 7). Belk [18], also differentiated between the definition of the sharing economy and collaboration consumption. His definition of SE involved "true sharing, entailing temporary access rather than ownership, no fees or compensation, and the use of digital platforms. Most of the commercial platforms included in the sharing economy not belong there" [18] (p. 1597). Moreover, the definition of collaboration consumption involved "people coordinating the acquisition and distribution of a resource for a fee or other compensation. By including other compensation." [18] (p. 1597). However, the definition of collaboration consumption is still related to the SE framework. Therefore, it could be considered one of the definitions of SE, as there is no consistent definition.

Ecosystems are becoming increasingly known and understood; new models are being developed to capture value [19]. The ecosystem has many definitions related to several elements. For example, Thomas and Ritala [20] defined it as a set of mutual understandings among ecosystem participants regarding the central, enduring, and distinctive characteristics of the ecosystem value proposition (p. 14). However, Wallace [21] employs an environmental understanding to define the ecosystem as "the point at which one or more humans consume the asset [of nature] is the point where the service occurs and should be evaluated" (p. 240). Danley and Widmark [22] identify ecosystem definitions from the service and conceptual prospects. They conclude that the Millennium Ecosystem Assessment configured a comprehensive definition of ecosystem services as "the benefits people obtain from ecosystems" [23] (p. V). Haines-Young and Potschin [24] adopt a definition in the same direction: "the outputs of ecosystems (whether natural, semi-natural or highly modified) that most directly affect the well-being of people" (p. 9).

In addition, Peltoniemi and Vuori [25] define the ecosystem as a "system of organisms occupying a habitat, together with those aspects of the physical environment with which they interact" (p. 2). Therefore, it is necessary to review several definitions to understand the meaning of the participants' terms and whether there are differences between researchers in reaching a definition. For example, Verba et al. [26] define participation as an "activity that is intended or has the consequence of affecting, either directly or indirectly, government action" (p. 7). In comparison, Park and Perry [27] identified participation as "individual and collective engagement in public affairs" (p. 191). However, nowadays,

participants work digitally, participating online. Lutz et al. [28] define online participation as "the creation and sharing of content on the Internet addressed at a specific audience and driven by a social purpose" (p. 881). In addition, Morozov and other auothers [29–31] considers online participants as affiliation businesses, not peer-to-peer businesses.

## 3. Review Methodology

According to Cheng [32], the review methodology followed the same guidelines. Table 1 shows 6 phases of the adapted method. The first phase is to determine the purpose of the study, the second phase is setting the search strategy to inform the search process for the review, and then followed by the third phase is the search strings by using keywords such as "sharing economy", "consumption of collaboration", "Participants", "Ecosystem", and "Digital technologies". Several articles related to SE were found during the initial research. A combination of keywords was used in the search to identify relevant studies about participants in SE platforms, such as "sharing economy AND Ecosystem", "participants AND Sharing Economy", and "sharing economy AND Review". The fourth phase is to use the above keywords to search in a database for articles with titles, abstracts, or keywords that contain these keywords. A search on Google Scholar and Scopus was performed, as well as investigations on ScienceDirect, JSTOR, Emerald, Elsevier, and Wiley. The fifth phase is the screening and inclusion criteria, and the sixth phase is the exclusion criteria. The total number of articles selected for further analysis was 70; 27 articles were related and used. Each is organised by the author, the year of publication, the theoretical lens, the data sources, and the main findings, and classifying the article based on the participant role (primary and secondary). The 27 papers provide insight into the relationship between the participants of the sharing economy ecosystem, in addition to 21 articles focusing on SE (definition) for the terms of SE, Participants, and the Ecosystem.

**Table 1.** Systematic review and protocols adopted.

| Review Phases | Description | Focus on the Review |
|---|---|---|
| Purpose | Aim of the literature review | To review the previous studies on the participants of the SE ecosystem |
| Search strategy | Plan to inform the search process for the review | Using keywords to search specified databases informed by screening and exclusion criteria |
| Search strings | Combination of keywords used to conduct the search for literature | "Sharing economy", "consumption of collaboration", "Participants", "Ecosystem" "Digital technologies". A combination of keywords was used in the search to identify relevant studies about participants in SE platforms, such as "sharing economy AND Ecosystem", "participants AND Sharing Economy," and "sharing economy AND Review." |
| Databases | Independent online database with citation data and indexes of scholarly writings | A search on Google Scholar and Scopus was performed, as well as investigations on ScienceDirect, JSTOR, Business Source Ultimate, Emerald, and Wiley. |
| Screening and inclusion criteria | Conditions for selecting and including review sources | The screening criteria for the review are as follows:<br>• Empirical and theoretical peer-reviewed journal articles<br>• Sharing economy studies<br>• Research on "participants" and "Sharing economy ecosystem" concepts and challenges |
| Exclusion criteria | Conditions for omitting publications during the review process | The exclusion criteria for the review are as follows:<br>• Duplicates<br>• Master's theses, doctoral dissertations, textbooks, and unpublished working papers<br>• Articles that use the term "Sharing Economy" and "Collaboration consumption" beyond the scope of participants of the SE Ecosystem criteria. |

### Summarises the Studies Reviewed on Participants in the SE Ecosystem

Table 2 classifies the participants in the SE ecosystem into primary and secondary participants. The classification is based on the connection to the core network/ecosystem

and the role of the participants in the ecosystem. The primary participants are subdivided into groups: customers are subdivided into New Customers (NC) and Current Customers (CC); providers into Product Providers (PP) and Service Providers (SP); and mediators are subdivided into Small and Medium Mediators (SMM) and Large Mediators (LM). The secondary participant is sub-grouped into governments into Local Government (NG) and National Governments (NG). Theories and methodologies within the academic literature on the sharing economy ecosystem are also examined.

**Table 2.** Summary of the studies reviewed on participants in the SE ecosystem.

| Author | Theoretical Lens | Data Sources | Key Findings | Primary Participants | Secondary Participants |
|---|---|---|---|---|---|
| Martin [9] | Socio-technical transitions, theory, and framing theory | Conceptual paper | Economic opportunity, sustainable consumption, and sustainability are three methods used by those seeking to empower the niche. | Mediators (LM) | Government (NG) |
| Cheng [32] | Tourism Theory Hospitality theory | Conceptual Paper | Researchers organise their research into five streams or clusters that help identify potential new directions | Customers (CC) | Mediators (N/A) |
| Hamari et al. [8] | Self-determination theory | Data collected from an online survey on customers' attitudes and collaboration consumption | The gap between attitude and behaviour is relatively small in comparison to the studies of technology adoption in general | Customers (CC) | N/A |
| Böcker and Meelen [33] | Self-determination theory. Hierarchical needs theory | Data collected from an online survey on sharing motivations | It is essential to differentiate between the various types of business in SE | Providers (SP) | N/A |
| Laurell and Sandström [7] | Institutional theory | Conceptual paper | Tensions create a state of instability related to SE as a contemporary phenomenon | Mediators (LM) | Governments (NG) |
| Lan et al. [34] | Social identity theory | Data collected from an in-depth analysis of real-life factors (a case study of MOBIK) and interviews focused on business value across the SE participants | A sustainable sharing business must identify and realise value co-creation behaviours in SE | Customers (CC) and Providers (SP) | Mediators (LM) |
| Miralles et al. [35] | Organisation theory | Data were collected from a comparative case study across 18 AFNs identifying five SE models of AFNs with unique shared resources and organisational mechanisms | Participants pool their resources, including production, marketing, and distribution | Mediators (SMM) | Providers (PP) |
| Ganapati and Reddick [36] | Economic theory | Conceptual paper | A harsher form of capitalism could be considered in the sharing practice | Government (LG) | Customers (CC) and Providers (SP) |
| Cheng et al. [12] | Expectancy-disconfirmation theory | Data were collected from 294 questionnaires from Chinese mobile car-hailing service providers | Competence, empathy, and information congruency are key in quality offline services. | Providers (SP) | N/A |
| Jin et al. [37] | Neo-Marxist theory | Conceptual paper | There is a close relationship between the digital divide and the intelligent city concept in SE | Providers (SP) | Customers (CC) |
| Sutherland and Jarrahi [6] | Design Theory | Conceptual paper | There are two types of SE organising models: centralised and decentralised | Customers (CC) | N/A |

**Table 2.** *Cont.*

| Author | Theoretical Lens | Data Sources | Key Findings | Primary Participants | Secondary Participants |
|---|---|---|---|---|---|
| Boons and Bocken [38] | Transition theory | Conceptual paper | Increasing the level of protection increases the chances of a successful niche in social engineering systems | Mediators (LM) | Providers (SP) |
| Mauri et al. [39] | Social theory; signalling theory | Conceptual paper | Sharing platform managers could reduce transactional uncertainty by helping sellers understand what additional features they should include in their profiles | Customers (CC) | Mediators (LM) |
| Ma et al. [40] | Self-determination theory and production theory | Data were collected from 50 stakeholder interviews with representatives of sharing mobility businesses in China | Changing consumption patterns and the growth of supply chains in the new SE | Mediators (SMM) | Government (NG) |
| Leunget al. [2] | Tourism theory. economic theory | Conceptual paper | SE should take steps to manage its external impacts through collaboration | Providers (SP) | Government (LG) |
| Eckhardt et al. [3] | Social theory and theory of social production | Conceptual paper | SE can be used to understand all facets of marketing, including consumer behaviour and culture | Customers (CC) | Providers (SP) |
| Simon and Roederer [41] | Lifespan theory and Self-determination theory | Data were collected from an online questionnaire with customers interested in flat sharing | The presence of other sharers directly and robustly impacts customer satisfaction in flat sharing | Customers (NC) | N/A |
| Pies et al. [4] | Social theory | Conceptual paper | Business models with SE hybridity face three challenges requiring managerial governance and communication abilities | Mediators (SMM) | Government (NG) |
| del Mar Alonso-Almeida et al. [42] | Social theory | Data collected from surveys conducted during various events with 384 postgraduate students regarding the level of consumer awareness | Through SE participation, consumers become more aware of their consumption habits, creating new materialism | Customers (CC) | N/A |
| Govindan et al. [43] | Iterative theory | Data were collected from in-depth interviews and workshops conducted with 38 industrial managers | The barriers to industrial SE can be attributed to a lack of trust and transparency, a lack of business models, or an absence of technology platforms | Mediators (SMM) | N/A |
| Hossain [44] | Diverse theories; self-determination theory, Economic theory | Conceptual paper | Difficult tasks are often involved in emulating SE firms | Mediators (LM) | Government (NG) |
| Sands et al. [45] | Social exchange theory self-determination theory | Conceptual paper | Provides an overview of the types of SE actors | Providers (SP) | Mediators (SMM) |
| Lim [5] | Marketing Theory | Conceptual paper | By enabling consumers to become producers, SE leads to greater competition | Providers (SP) | Customers (CC) |
| Song et al. [46] | Economic theory | Conceptual paper | Peer-to-peer trading has more economic advantages than pure producers and consumer models | Providers (PP) | Customers (NC) |

**Table 2.** *Cont.*

| Author | Theoretical Lens | Data Sources | Key Findings | Primary Participants | Secondary Participants |
|---|---|---|---|---|---|
| Shen et al. [47] | Social exchange theory | Conceptual paper | Prosumers are now being considered when evaluating brand value in SE | Providers (SP) | Mediators (LM) |
| Pereira and Silva [48] | Institutional theory | Data were collected from seven interviews with public and private agents (socio-technical actors) | There is a potential conflict of interest between public and private agents as a consequence of the integration of these several initiatives | Mediators (LM) | Government (NG) |

## 4. Results

This review examined valuable insight into the participants of the SE ecosystem. It classifies the participants in the SE ecosystem into primary and secondary participants. The classification is based on the connection to the core network/ecosystem and the role of the participants in the ecosystem.

### 4.1. Primary Participants vs. Secondary Participants

The primary participants are subdivided into groups: customers are subdivided into New Customers (NC) and Current Customers (CC); providers into Product Providers (PP) and Service Providers (SP); and mediators are subdivided into Small and Medium Mediators (SMM) and Large Mediators (LM). The secondary participant is sub-grouped into local government (NG) and National Government (NG).

#### 4.1.1. Customers (Primary)

Customers are the beneficiaries of the products or services from producers or providers through mediators. The review subdivides customers into New Customers (NC) and Current Customers (CC). As shown in Table 2, customers are among the primary participants in the SE ecosystem, according to most of the reviewed articles. Furthermore, Current Customers (CC) were associated with the National Government (NG) and the Local Government (LG) [36]. In contrast, NC is not associated with any government in the reviewed articles. Furthermore, NC was linked only to Product Providers (PP) [46], while CC was attached only to the Service Providers (SP) [5,34,37]. Moreover, CC was connected to both the Large Mediators (LM) and the Small and Medium Mediators (SMM) [34].

#### 4.1.2. Mediators (Primary)

Mediators are the platforms that mediate the provider's services or products. The review subdivides the mediators into Large Mediators (LM) and Small and Medium Mediators (SMM). As shown in Table 2, the mediators were the primary participants in the SE ecosystem in most of the reviewed articles [9,44]. Furthermore, the LM and the SMM were only associated with the National Government (NG) [4,7,9,40]. In addition, the LM was only linked to the Service Providers (SP) [34,38]. In contrast, the SMM was related to both Service Providers (SP) [45] and Product Providers (PP) [35].

#### 4.1.3. Providers (Primary)

Providers are the producers of products or provide services to customers. Therefore, the review subdivides them into Service Providers (SP) and Product Providers (PP). As can be seen in Table 2, the providers were the primary participants in the SE ecosystem in most of the reviewed articles. Furthermore, the SP were the leading group in the reviewed papers and was mainly linked to the LM and SMM [34,38,45]. In contrast, the PP were only connected to the SMM [35]. Moreover, they were associated with NC [46]. Furthermore, the SP were only related to the CC [5,34,37] and the LG [2].

### 4.1.4. Government (Secondary)

The government is the authority that manages the relations between the ecosystem participants through regulations to organise and protect their commitments. Therefore, the review subdivided government into the National Government (NG) and Local Government (LG). As shown in Table 2, the government was a secondary participant in the SE ecosystem in most of the reviewed articles. Furthermore, the NG was the leading group in the reviewed papers and was mainly linked to the LM and the SMM [4,7,9,40]. In addition, the NG and the LG were linked to the CC [36]. At the same time, LG was only connected to the SP [36].

In addition to the above findings, others are related to the SE ecosystem participants and their relevance, including different elements of all the marketing and management domains, such as consumer behaviour, empirical modelling, and strategy [3]. These elements are linked to the different business models, characterised by specific economic, social, and environmental perspectives, resulting from the value creation adopted by SE initiatives [4]. Furthermore, development in SE platforms is essential to achieve sustainable growth and compromise through the required sacrifice of business profits. However, SE leads to more sustainability in the concept patterns of sharing industries [5]. Sutherland and Jararhi [6] focused on technology as a reliable tool to connect the participants of the SE ecosystem. They highlighted the significant variation in the technologies used from one field to another on SE platforms. Technologies play an essential role in the SE ecosystem. "The technologies studied under the sharing economy vary significantly, from ride-sharing services to distributed currencies to freelancing platforms. Research perspectives vary similarly, including tourism, governance, design and digital gig work" [6] (p. 24). Moreover, Laurell and Sandström [7] discuss the market and non-market logic tensions.

Furthermore, many studies considering the technological impact perspectives are due to SE research variation across many fields, providing a better conceptualisation of SE technologies and mediation [8,9]. Therefore, sharing is considered a distinctive consumption preference [13]. Furthermore, online business and mobile applications help to facilitate modern SE transactions driven by the sharing purpose [8]. The review also provides valuable insights into the study participants' roles. First, it divides the functions into primary and secondary groups inside the same ecosystem, which is helpful in understanding the participants' impacts through the previous studies, as it shows that the mediators, customers, and providers are primary participants inside the SE ecosystem. At the same time, the government is a secondary participant in the ecosystem.

Generally, the main findings are that the participants have a common focal point: they work under the concept of the system's organisms. Moreover, interactions between all participants in the SE ecosystem improve the sharing business's performance and sustainability [49]. In the summaries of their main findings, most of the previous literature has focused on different participants in the SE ecosystem. However, in these studies, the ecosystem participants were not fully integrated, although sharing business remains more sustainable, and sharing still leads to more sustainability related to the ability of platforms.

### 4.2. Methods Used in the Reviewed Articles

The methods used in the reviewed articles were varied, although conceptual papers constituted 32% of the overall reviews. Interviews constituted 29%; online surveys 21%; and online questionnaires and case studies 13%. In addition, 68% of the studies in the review adopt a non-empirical approach versus 32% that adopt the empirical approach, as shown in Figure 1.

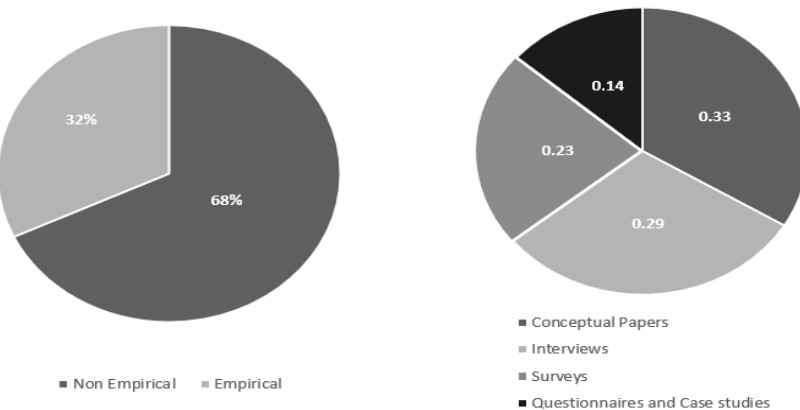

**Figure 1.** Methods used in the reviewed articles.

### 4.3. Theories Used in the Reviewed Articles

This section highlights and explains the most frequently used theories related to the study participants. Furthermore, Table 3 presents previous studies' most frequent theories and examples. First, the social theory focuses on the different challenges for social business. An example of the use of theory is Pies et al. [4]. They used this theory to explain the challenges of hybrid business models. Moreover, del Mar Alonso-Almeida et al. [42] presented the social theory in their study to clarify how customers became more aware of consuming habits. The second theory in Table 3 is the self-determination theory which is focused on participation and is not linked directly to sustainability unless positive attitudes also accompany it. An example of the use of a view is Böcker and Meelen [33]. They used this theory to explain the essential differences between the various business types in SE. Sands et al. [45] and Andonopoulos et al. [45] refer to this theory to explain the providers and mediator relationship. The third and final theory in Table 3 is the economic theory. It explains the lack of regulation and policies impacting SE business. Song et al. [46] used this theory to explain why peer-to-peer trading has more economic advantages.

**Table 3.** Most frequently used theories.

| Theory | Number of Articles | Explanation | Examples of Previous Studies |
|---|---|---|---|
| Social theory | 5 | Different challenges for social business. | Mediators (SMM) and Government (NG): Pies et al. [4] Customers (CC): del Mar Alonso-Almeida et al. [42] |
| Self-determination theory | 4 | Participation is not linked directly to sustainability unless positive attitudes also accompany it. | Providers (SP): Böcker and Meelen [33] Providers and Mediators: Sands et al. [45] Customers (CC): Hamari et al. [8]. |
| Economic theory | 4 | The lack of regulation and policies impacts SE business. | Providers (PP) and Customers (NC): Song et al. [46] Mediators (LM) and Government (NG): Hossain [44] |

### 4.4. Consequences of the Technologies Value on the Participants

In addition, the framework in Figure 2 highlights the consequences for participants. It explains the value of the technologies presented in Table 4. Starting with the mediators as they obtain value from the technologies as they improve business sustainability. Next, the providers value the technologies as they support engagement with market needs. Finally, the

customers obtain value from the technologies as they increase customer satisfaction. Finally, the government obtains value from the technologies as they enhance the economy's performance.

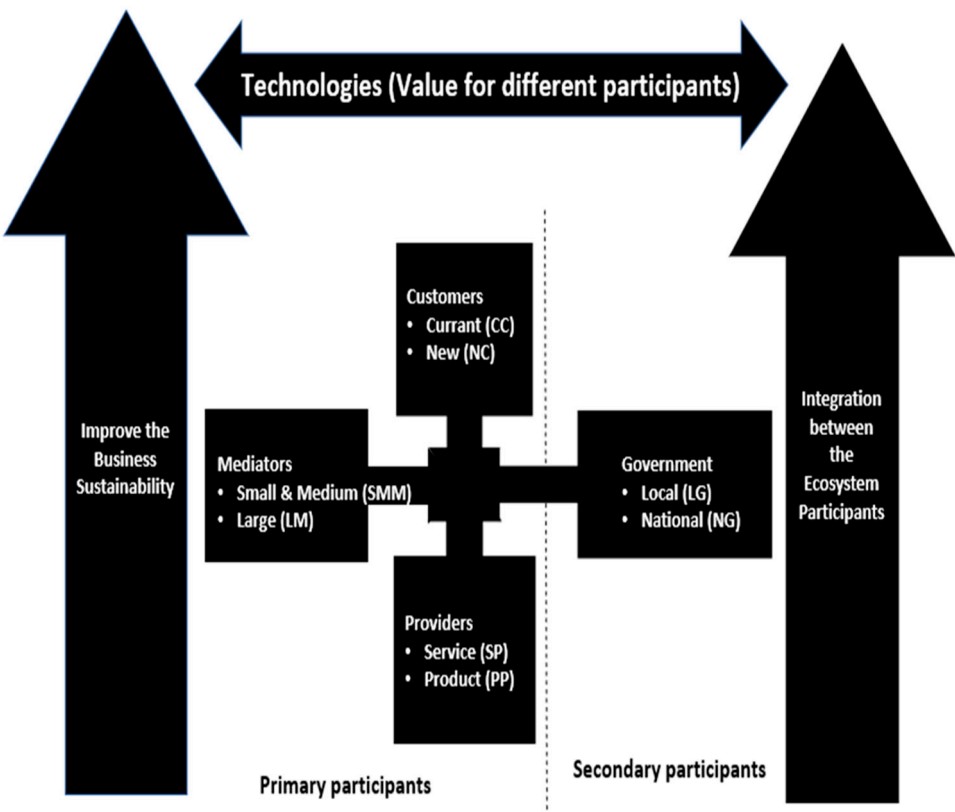

**Figure 2.** An integrated framework of the sharing economy ecosystem.

**Table 4.** Consequences of the technologies value on the Participants.

| Participants | Consequences |
| --- | --- |
| Mediators | Mediators obtain value from the technologies:<br>• Improve the business sustainability<br>• Growth of the business revenue<br>• Adopt the innovative business model |
| Providers | Providers obtain value from the technologies:<br>• Support engagement with market needs<br>• Improve the market standers due to the competition |
| Customers | Customers obtain value from the technologies<br>• Increase customer satisfaction<br>• Improve the quality and the prices of the products and services<br>• Provides varieties of services and products |
| Government | Government obtains value from the technologies<br>• Enhance the economy's performance<br>• Enhance the government's services<br>• Enhance the government's influence |

## 5. Discussion

The research has two objectives. The first objective is to understand the value of technologies in integrating SE participants and their impact on business sustainability. The second objective is to clarify the role of the participants in the SE ecosystem by subdividing

them into groups and developing a framework to explain the primary and secondary roles they play in each group. The findings indicate that the primary participants are mediators, providers, and customers, and the secondary participant is the government. In addition, the framework presented in Figure 2 explains the needs for the technologies in the SE participants. Therefore, SE has grown as an alternative to traditional business forms of ownership, provided by digital entities that allow users to connect [50].

Consequently, markets can be created that compete with traditional businesses [51]. Furthermore, the social factor in sharing business adds flexibility and freedom to SE models [18]. Again, digital transformation and the increase in the use of the internet help to mediate transactions between providers and customers [52], increase trust between strangers [51], and significantly reduce the cost of transactions [53]. Although, many kinds of research focus on Service Providers and consumers in the SE ecosystem and on several topics related to these participants. For instance, the challenges they face, including the impact of the trust-based concept on commercial sharing [54] and how this could be treated by improving business communication. The research also considered how organisations understand their need to innovate their SE business model, which is essential for their continuity and to cope with SE market changes. In addition, the recognition of the debate around several topics related to SE mainly explored the role of the SE participants. For example, recent research has mentioned digital technologies in the SE ecosystem and how the SE impacts the economic and social aspects [32]. The researchers provide deep insights into the value of actors in the SE ecosystems, which is a significant element in the sharing practice. Nevertheless, using digital platforms often requires contributions from various parties [55]. Moreover, it represents a fundamental challenge inside the SE platforms [48]. It is crucial, considering that the actors are empowered to establish marketplaces even in the smallest communities and change the price policy in a specific industry. However, they need to integrate their management strategy with the dynamic nature of SE, not by creating a direct value [56]. Furthermore, actors in SE will lead to more sustainability [45], which is a significant element in any business model. As a whole, the outcomes of an ecosystem depend directly on the participants' behaviour. Lutz et al. [57] agree with Morozov [29] and claim that online participants are quite passive. In addition, Andreotti et al. [30] focus on participants" behaviour and claim that "personal values and attitudes can be assumed to affect the relationship be-tween motives and (non-)participatory behavior. Again, socio-economic variables may be associated with distinct attitudes or value sets" (p. 15).

## 5.1. Theoretical Implications

The theoretical implications of this study highlighted the value of integrating the different SE theories, digital sharing, and participant behaviour. In addition, it classifies the participants into primary and secondary participants. It divides them into groups to help better understand their role in the SE ecosystem, leading to the potential development of the current theories. This study helps understand the SE ecosystem participants and their impact on their interactions. The research builds a stronger foundation between theories and practice. The review supported institutional theory (the discursive institutionalist approach). The review contributes to the theory by employing the theoretical perspective used in the reviewed articles on the participants in the SE ecosystem. Furthermore, it presents and identifies the knowledge by extending the findings and the discussions of the review into a holistic view of the participants. The research also contributes by understanding the impact of the integration of the participants on the business model. As a whole, the outcomes of an ecosystem depend directly on the participants' behaviour.

## 5.2. Managerial Implications

From a managerial standpoint, the analysis of the elements of the SE ecosystem provides insights into dealing with the sector. The review has considered the current needs of the SE business, covering many sectors, such as hospitality, transportation, information, education, food, energy, and fashion. Furthermore, SE and collaborative consumption

were shown to be connected. They digitally support businesses' connection with the communities, which have grown exponentially in recent years. The growth of digital use helps to explore further opportunities and integration with SE businesses, which could lead to massive changes in the services and goods provided by the participants in the SE ecosystem [58–61].

Nevertheless, by exploring the roles of participants at multiple levels of the SE ecosystem, this study sheds light on developing new business models that can significantly impact SE businesses in both their current and new contexts [62–64]. It is beneficial for governments to create a regulatory atmosphere that fosters a more friendly business environment to create conditions for the growth of businesses in order to create more job opportunities. Furthermore, the study enhanced the understanding of the needs of small and medium businesses and the expectations of governments. Moreover, governments played a secondary role in the SE ecosystem. Therefore, governments must strengthen, develop, and enforce small and medium businesses to play a primary role in the SE ecosystem.

Accordingly, connectivity has increased, with the commercialisation of ownership and the agency of technology identified as the main trends. While the SE concept has been long-established, digital technologies in this field have only recently been developed. Many SE researchers have discussed the SE ecosystem and how it is more efficient in generating new business and opens up many networks to consumers, providing them with needed goods or services. Furthermore, some recent studies have explored and explained the roles of the participants in SE, which has helped to increase business size, which is predicted to grow by up to USD 3.5 billion by 2025 [5,65]. Therefore, this paper intends to contribute by providing a condensed review of the SE ecosystem to explain the future impact of new social and economic configurations and the development of the SE platform and business model elements.

## 6. Conclusions

We conclude that classifying participants into primary and secondary helps business leaders understand each participant's current role and develop the secondary participants in the ecosystem to improve SE business in the future. Furthermore, the subdivision of the participants into groups sheds light on each participant type. In addition, the improvement in digital technologies positively reflects the integration between the participants in the SE ecosystem, which leads to more business sustainability. As shown in Figure 2, a framework is suggested that explains the value of the technologies to improve the integration between the participants and the impact on business sustainability. Moreover, the framework describes the role of the participants inside the SE ecosystem by classifying them as primary and secondary and subdividing them into groups to provide a clear understanding of which groups are considered in previous studies, as shown in Figure 2. More insights are needed into these types to understand how specific groups perform in SE and develop strategies according to SE business needs.

## 7. Directions for Future Studies

This paper found answers for a few future studies recommended in some reviewed articles. As shown in Table 5, some key themes from the review were explored. However, there remain unanswered questions. The table highlights the key themes in the thought, what has been learned from it, and what we still need to know.

**Table 5.** Key themes and directions for future studies.

| Participants | What We Know | What We Need to Know |
|---|---|---|
| Mediators | Integrating the private and public sectors contributes to the stability of SE. | Could the integration between the private and public sectors affect the efficiency and cost of the sharing practice? [48] |
| Providers | Prosumers are more motivated to share business than providers. | Advanced technology impacts the services providers' business positively (increases the awareness of their business) or negatively (increases the value of negative online reviews) [12]. |
| Customers | Current customers are braver in exploring services and products than new ones. | What technology could impact the new customers of the sharing business? [8] |
| Government | The government's role in the SE ecosystem is as a secondary participant. | Can policymakers help to enhance the influence of the government in becoming a primary participant in the SE ecosystem? [36] |

This study highlighted that it is worth considering the value of the technologies in integrating the participants in the SE ecosystem, together with business performance and sustainability. This research helps to highlight the significant influence of digital technologies on the integration between participants in the studies conducted in the last five years. Based on the review, the theoretical and managerial implications focused on the value that returns to the business from integrating the participants in the SE ecosystem. There is, however, no evidence for this point. Therefore, it would be helpful for future research to consider more values of the integration of the SE participants.

**Author Contributions:** Conceptualization, S.A., J.A.-A. and A.B.; methodology, S.A. and J.A.-A.; formal analysis, S.A., J.A.-A. and A.B.; investigation, S.A., J.A.-A. and A.B.; data curation, S.A., J.A.-A. and A.B.; writing—original draft preparation, S.A. and J.A.-A.; writing—review and editing, S.A., J.A.-A. and A.B.; visualization, S.A., J.A.-A. and A.B.; project administration, S.A. All authors have read and agreed to the published version of the manuscript.

**Funding:** This research received no external funding.

**Institutional Review Board Statement:** Not applicable.

**Informed Consent Statement:** Not applicable.

**Data Availability Statement:** Data is contained within the article.

**Conflicts of Interest:** The authors declare that they have no known competing financial interests or personal relationships that could have appeared to influence the work reported in this paper.

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
