# Peer review of "Sharing-Economy Ecosystem: A Comprehensive Review and Future Research Directions"

_sustainability, doi:10.3390/su15032145_

Round 1

Reviewer 1 Report

The work is very interesting. From the introduction the reader understands the reason for the research and the contribution to knowledge.

I recommend dividing the first section into introduction and literature so as to have a more orderly reading. The section "who are the participants..." can be included in the literature

The last paragraph of the introduction (contribution) should be more schematic.

The reference to the beginning of section 2 is essential. I had trouble tracking her down. The reference doesn't seem right to me. Is it perhaps the 32? (Cheng, M., 2016. Sharing economy: A review and agenda for future research. International Journal of Hospitality Management, 486 57, pp.60-70. )?

The discussion is very well done and interesting. some points should be taken up again in the conclusion. The conclusion section is too short considering the size of the other sections and should be at the end of the work. The future research section is too short and can be incorporated into discussions.

Minor comments.The introduction could be shortened. Better when the introduction is short and immediately invites you to read. (for example: bottom Page 2 too many passages "For example" ). 

Perhaps these papers could be cited:

Curtis, S.K., & Lehner, M. (2019). Defining the sharing economy for sustainability. Sustainability, 11(3), 567.

Author Response

Response to Reviewer 1 Comments

The work is very interesting. From the introduction the reader understands the reason for the research and the contribution to knowledge. I recommend dividing the first section into introduction and literature so as to have a more orderly reading. The section "who are the participants..." can be included in the literature.

Response (R): Many thanks for your comments. Following your suggestions, the introduction section has been divided to include the “who are the participants? “Section.

The last paragraph of the introduction (contribution) should be more schematic. The reference to the beginning of section 2 is essential. I had trouble tracking her down. The reference doesn't seem right to me. Is it perhaps the 32? (Cheng, M., 2016. Sharing economy: A review and agenda for future research. International Journal of Hospitality Management, 486 57, pp.60-70.)?

R: Many thanks for your comments. Consequently, the manuscript has been revised to address the shortcomings identified. The reference has been corrected.

The discussion is very well done and interesting. some points should be taken up again in the conclusion. The conclusion section is too short considering the size of the other sections and should be at the end of the work. The future research section is too short and can be incorporated into discussions.

R: Following your suggestions the part from the discussion section moved to the conclusion section. The conclusion and future research sections have been expanded. However, regarding the future study it meant to be classified by participants in order to provide a clear direction for the future research.

Minor comments. The introduction could be shortened. Better when the introduction is short and immediately invites you to read. (for example: bottom Page 2 too many passages "For example”). Perhaps these papers could be cited: Curtis, S.K., & Lehner, M. (2019). Defining the sharing economy for sustainability. Sustainability, 11(3), 567.

R: The introduction section is shortened (as it was divided into sections), and the paper has been cited.

Reviewer 2 Report

Dear Authors,

Your article is quite interesting and well written. However, it has several spelling mistakes, por example, in table one: current CC is mentioned as currant CC, and so on.

I think you should consider also Web of Science (and Clarivate) journals (in addition to Scopus), as it is also a major indexing means. Your list mentions Emerald, etc, but not Elsevier, Taylor and Francis, and a few other major publishers of research articles. Why did you exclude/ignore them? Probably some of your conclusions could change, if you include articles published in these journals.

Author Response

Response to Reviewer 2 Comments

Your article is quite interesting and well written. However, it has several spelling mistakes, for example, in table one: current CC is mentioned as currant CC, and so on.

R: Many thanks for your comments and the opportunity to revise the manuscript. The entire manuscript has been overhauled to address the shortcomings identified.

I think you should consider also Web of Science (and Clarivate) journals (in addition to Scopus), as it is also a major indexing means. Your list mentions Emerald, etc, but not Elsevier, Taylor and Francis, and a few other major publishers of research articles. Why did you exclude/ignore them? Probably some of your conclusions could change, if you include articles published in these journals.

R: Following your suggestions, Elsevier has been added to the publishers. Most of the reviewed articles that were reviewed in Google scholar and Scopus were listed in the Web of Science as well. Accordingly, there are no changes at the findings and the conclusion sections

Reviewer 3 Report

The study is interesting and contributes to a better understanding of the sharing-economy ecosystem. However, it has several methodological weaknesses and some organizational problems. On the other hand, it has virtually no practical implications for participants in the ecosystem.

On pg. 2 lines 78 to 104 – the authors present theoretical contributions. I think this should be considered in the conclusions section. Review

Methodology: How was the Synthesis and analysis of records done?

How do authors ensure the reliability of the review process?

In the findings: it would be important to highlight aspects such as: Number of articles per year analyzed and main journals.

Managerial implications add very little. It will be important to show how this study can bring practical contributions.

The theoretical and managerial implications, as well as study limitations and directions for future research, should be integrated into the conclusions section.

Conclusions must present the study limitations.

Author Response

Response to Reviewer 3 Comments

The study is interesting and contributes to a better understanding of the sharing-economy ecosystem. However, it has several methodological weaknesses and some organizational problems. On the other hand, it has virtually no practical implications for participants in the ecosystem. On pg. 2 lines 78 to 104 – the authors present theoretical contributions. I think this should be considered in the conclusions section. Review

R: Many thanks for your comments. We have now developed and incorporated practical implications.  The change on the theoretical contributions has been made.

Methodology: How was the Synthesis and analysis of records done? How do authors ensure the reliability of the review process? In the findings: it would be important to highlight aspects such as: Number of articles per year analysed and main journals.

R: Following your comments, the authors adopt six phase to ensure the reliability of the review process., The first phase is to determine the purpose of the study, the second phase is setting the search strategy to inform the search process for the review, and then followed by the third phase is the search strings by using keywords such as "sharing economy", "consumption of collaboration". A combination of keywords was used in the search to identify relevant studies about participants in SE platforms, such as "sharing economy AND Ecosystem," The fourth phase is to use the above keywords to search in a database for articles with titles, abstracts, or keywords that contain these keywords. A search on Google Scholar and Scopus was performed. The fifth phase is the screening and inclusion criteria, and the sixth phase is the exclusion criteria. The total number of articles selected for further analysis was 70; 27 articles were related and used. Each is organised by the author, the year of publication, the theoretical lens, the data sources, and the main findings, and classifying the article based on the participant role (primary and secondary).

Managerial implications add very little. It will be important to show how this study can bring practical contributions.

R: Many thanks for these observations. Following your comments, the managerial implication has been revised.

The theoretical and managerial implications, as well as study limitations and directions for future research, should be integrated into the conclusions section. Conclusions must present the study limitations.

R: As indicated previously, the conclusion and the managerial implications section has been revised, and regarding the future study it meant to be classified by participants in order to provide a clear direction for the future research. Consequently, the manuscript has been revised to address the shortcomings identified. Many thanks for using your considerable understanding and awareness of literature to help shape the key arguments. We hope these efforts have allayed all your concerns.  

Round 2

Reviewer 2 Report

Dear Authors,

I think your paper looks great now. Congratulations!

Reviewer 3 Report

These authors have done significant improvements in this paper and, in my opinion, they have achieved a publishable version.